# Step-wise Sensitivity Analysis: Identifying Partially Distributed Representations for Interpretable Deep Learning

## Abstract

In this paper, we introduce a novel method, called step-wise sensitivity analysis, which makes three contributions towards increasing the interpretability of Deep Neural Networks (DNNs). First, we are the first to suggest a methodology that aggregates results across input stimuli to gain model-centric results. Second, we linearly approximate the neuron activation and propose to use the outlier weights to identify distributed code. Third, our method constructs a dependency graph of the relevant neurons across the network to gain fine-grained understanding of the nature and interactions of DNN's internal features. The dependency graph illustrates shared subgraphs that generalise across 10 classes and can be clustered into semantically related groups. This is the first step towards building decision trees as an interpretation of learned representations.

## 1 Introduction

Deep Neural Networks (DNNs) have impressed the scientific community because of their performance and the variety of domains to which they can be applied. However, DNNs are difficult to interpret because of their highly complex non-linear and interconnected nature. The lack of transparency is a threefold problem. First, it inhibits adoption, especially in industries under heavy regulation and with a high cost of errors. Second, it makes it difficult to debug existing models and hampers development progress. Third, it prevents us from utilising the insights gained from the models for further knowledge discovery.

DNN interpretability is loosely defined, and it is also referred to as Explanatory AI, Understandable Machine Learning, and Deep Visualisation. DNN interpretability can be gained from a human-interpretable explanation of the reasons behind the network's choice of output (Ribeiro et al., 2016; Doshi-Velez & Kim, 2017). In a DNN the basis for a decision is encoded in features either as one neuron – local representation; or as a set of neurons – partially-distributed representation (PDR) (Li et al., 2016; Fong & Vedaldi, 2018).

The identification of PDRs and their interactions remains the main hindrance to end-to-end interpretability systems (Olah et al., 2018). Once identified, PDRs will enable us to give much finer-grained explanations (e.g., an image is a shark because the network detected a sea, sharp teeth, a long fin, etc.). In this paper, we introduce our novel technique, step-wise sensitivity analysis (SSA), with which we make 3 contributions towards this vision: SSA 1) identifies PDRs; 2) illustrates interactions between related PDRs; 3) applies a novel perspective for interpretability – statistics across multiple input stimuli (instance-specific) to gain a general understanding of the DNN operation (model-centric).

The method produces *statistical topological interpretability*. That is, we analyse the network's properties as a directed graph over various inputs to produce a dependency graph between neurons. The dependency graph highlights the relationships between adjacent layers that are pertinent to the decision, and how these relationships are formed in each layer and across all layers to form a feature representation.

The remainder of this paper is organised as follows: Section 2 reviews related work; in Section 3 we introduce our technique for building the cross-input layer-wise dependency graph; Section 4

illustrates three novel neuron relevance metrics stemming from the dependency graph and how they can be used to determine and interpret neurons of interest; Section 5 makes concluding remarks and suggests how our work can be applied and extended.

## 2 BACKGROUND

In this section, we organise the existing effort dedicated to DNN interpretability, based on its three main limitations: functional vs. topological, instance-specific versus model-centric, single neuron versus layer analysis. We then compare step-wise sensitivity analysis with other works that address the same limitations.

First, recent attempts focus primarily on the input-output relationship, considering the network as a black-box function. These methods are classified as *functional* since they treat the entire network as a black-box function with an input, output and parameters, while the topological approach considers the structure of the network. One of the most investigated areas in the functional vein is sensitivity analysis, which produces a heatmap, illustrating the input parts relevant to the output decision. Examples include deconvolution (Zeiler & Fergus, 2014), sensitivity image-specific class saliency visualisations (Simonyan et al., 2013), guided-back propagation (Springenberg et al., 2014), and predictive difference analysis (Zintgraf et al., 2017). In contrast to these functional methods, Section 3 demonstrates how the sensitivity analysis technique can be modified to gain more granular information across layers.

The second limitation is that there are two completely opposite types of visualisation approaches for interpretability – *model-centric* or *instance-specific*. The model-centric approaches, such as activation maximisation (Erhan et al., 2009) and inversion (Mahendran & Vedaldi, 2015), synthesise a prototype image to explain a neuron. This can be generalised to every data point, but does not contain enough details to reason about particular mistakes or edge-cases. On the other hand, instance-specific methods operate on the level of a single instance, but this fine-granularity cannot be used to elicit principles applicable across a wider set of instances (e.g., the relevance of regions in a single image). Our methodology iterates over instance-specific results, and we compare the results for different classes to illustrate similar "thought patterns", that is, the relevance of filters is shared across classes. Thus, our approach can be viewed as both, instance-specific and model-centric.

Third, current approaches either explore a single neuron in isolation, such as sensitivity analysis and activation maximisation, or the entire layer, such as inversion (Mahendran & Vedaldi, 2015). The former approach assumes purely local representations, while the latter assumes fully-distributed representations. However, recent findings (Li et al., 2016; Fong & Vedaldi, 2018; Agrawal et al., 2014; Bau et al., 2017) suggest that every layer consists of a mixture of local and *partially-distributed* representations. Our approach addresses the last two limitations, in particular, it interprets the internal DNN structure across layers of the network to identify the different types of representations in each layer.

### 2.1 RELATED WORK

#### 2.1.1 RELEVANCE SCORE TECHNIQUES

Net2Vec (Fong & Vedaldi, 2018) builds on network dissection (Bau et al., 2017) to propose a method for selecting and combining relevant filters, and providing an explanation for filters in conjunction with each other, thus identifying and interpreting PDRs. This method determines the relevance of neurons by optimising the combinations of filters for classification and segmentation on proxy ad hoc tasks. In contrast, our method can ascertain the neuron relevance using the original data set, which makes it more generalisable as it is not necessary to compile explanatory datasets for various problems. On the other hand, similarly to our approach, in Landecker et al. (2013) it is argued that computing the relevance in a layer-by-layer fashion yields more fine-grained results, and draws the distinction between *functional* and *topological* approaches. In Bach et al. (2015) this idea is incorporated into a sensitivity analysis technique - Layer-wise-relevance propagation (LRP). Deep Taylor Decomposition (Montavon et al., 2017) generalises the output of such sensitivity analysis techniques to an importance score, which is computed in a topological layer-wise fashion. DeepLift (Shrikumar et al., 2017) proposes an alternative functional approximation method for computing the relevance score by comparing the activation of a neuron to a reference one. Our work is similar to the rel-

evance score techniques in that it uses the original sensitivity analysis approach as an importance score metric. In addition and similarly to the topological approaches, we suggest computing the importance score at each layer. However, we propose that the network's decision is driven by a partially-distributed representation rather than the entire layer. Hence, instead of distributing the relevance across all neurons, we only redistribute the relevance to a small number of outliers.

### 2.1.2 CONSTRAINED REDISTRIBUTION

Our work is comparable to excitation backpropagation (Zhang et al., 2016), which distributes the relevance to a subset of neurons. There are two important differences. First, excitation backpropagation focuses on improving the heatmap quality, while we investigate how to discover PDRs. Second, while excitation backpropagation uses a probabilistic winner-take-all sampling approach that is limited to neurons with ReLU activations and positive weight connections between adjacent layers, we deploy a more generalisable linear Taylor approximation and statistical analysis over multiple inputs to restrict the relevant neurons.

### 2.1.3 VISUALISATION

Our method resembles the approach in (Liu et al., 2017) in that we also generate a DAG and augment it with additional visualisations to produce an explanation. The main difference resides in that they use clustering to reduce the visual clutter, and group neurons together. In contrast, we propose a novel method to select only the relevant paths for investigation.

## 3 STEP-WISE SENSITIVITY ANALYSIS

Our interpretability method is based on the *sensitivity analysis* technique by Baehrens et al. (2010) that was first applied to DCNNs by Simonyan et al. (2013) to produce image-specific class saliency visualisations. Formally, given an image $\mathbf{l}_0$, a representation function $\mathbf{\Phi} : \mathbb{R}^{H \times W \times C} \to \mathbb{R}^d$ such that $\mathbf{\Phi}(\mathbf{l}) = \mathbf{o}$, and a neuron $n$ – approximate the activation of $\mathbf{o}_n$ with a linear function. In the neighbourhood of $\mathbf{l}_i$, this is achieved by computing the first-order Taylor expansion:

$$\mathbf{o}_n = \mathbf{\Phi}_n(\mathbf{l}) \approx \boldsymbol{\omega}^T \mathbf{l} + b \tag{1}$$

where $\boldsymbol{\omega}$ is the gradient of $\mathbf{\Phi}_n$ with respect to an image $\mathbf{l}$. The function is evaluated at image $\mathbf{l}_i$:

$$\boldsymbol{\omega} = \frac{\partial \mathbf{\Phi}_n}{\partial \mathbf{l}}\bigg|_{\mathbf{l}_i} \tag{2}$$

This formulation allows us to interpret the magnitude of the values of $\boldsymbol{\omega}$ as an importance metric corresponding to each pixel. In other words, these values indicate which *pixels* need to be changed the least to change $\mathbf{\Phi}(\mathbf{l})$ such that $\mathbf{o}_n$ (corresponding to a classification decision) is increased the most.

### 3.1 SINGLE BACK-PROPAGATION STEP

Sensitivity analysis performs a complete back-propagation pass to compute $\boldsymbol{\omega}$ in equation 2. The end result is a class saliency map, which is particularly useful to identify the image regions most pertinent to the network's decision. However, this is a very coarse-grained explanation since it only analyses the input-output relationship. There is no information regarding the effect of particular layers and neurons on the selection of the relevant image regions. We propose a much more fine-grained analysis based on the simple hypothesis that sensitivity analysis can be used in an analogous way to determine the relevance between adjacent layers.

Instead of trying to approximate $\mathbf{o}_n$ directly, we consider $\mathbf{\Phi}$ to be defined as the successive composition of smaller functions that represent the transformations of data between layers:

$$\begin{aligned} \mathbf{\Phi}(\mathbf{l}) &= f^l(\mathbf{\Phi}^{l-1}(\mathbf{l})) \\ &= f^l \circ f^{l-1} \circ f^{l-2} ... \circ f^1(\mathbf{l}) \end{aligned} \tag{3}$$

where $l = 1...L$, $L$ is the network's depth, and each layer denoted as $f^l : \mathbb{R}^{d'} \to \mathbb{R}^d$ represents the operation applied by layer $l$, when $d'$ is the output dimensionality of the input layer $f^{l-1}$ and $d$ is

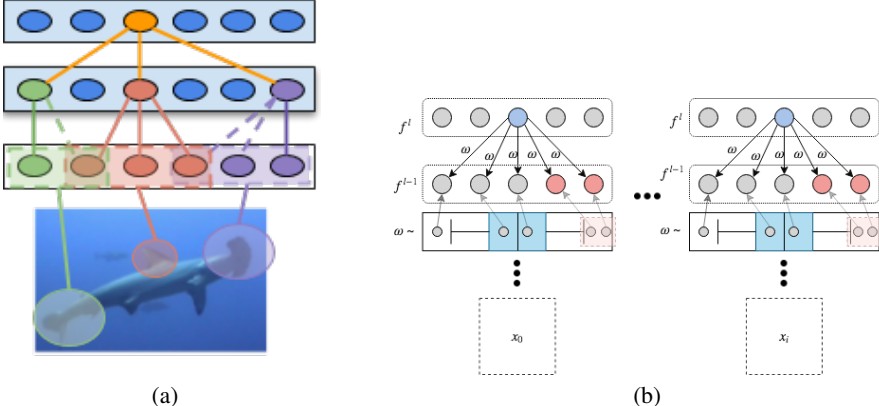

(a)           (b)

Figure 1: (a) A sketch of how Step-wise Sensitivity Analysis can be used to provide interpretation for a shark prediction (for actual output examples, see Fig. 3). Each step identifies PDRs of relevant neurons, and for each PDR recursively traverses downwards. (b) Schematic Representation of Step-wise sensitivity analysis representing the two novelties in the single step between adjacent layers. First, the relevant neurons are the ones with positive outlier values in the distribution of $\boldsymbol{\omega}$, represented with a boxplot ($\boldsymbol{\omega} \sim$). Second, the analysis is aggregated across instance-specific inputs to gain model-centric results.

the output dimensionality of layer $l$. Starting with neuron $n$ from the top layer $f_n^l$, we conduct only the last step of back-propagation to compute the $\boldsymbol{\omega}^{l-1} \in \mathbb{R}^{d'}$ values, where the node of the network under consideration is $f^{l-1} \in \mathbb{R}^{d'}$. We call our approach *Step-wise sensitivity analysis* (SSA) since it iteratively applies sensitivity analysis through the computation of a single back-propagation step. The result of a single back-propagation step can be seen as performing one step of step-wise sensitivity analysis between a higher layer $l$ and a lower one $j$:

$$\boldsymbol{\omega}_{n,i,:}^l = \left. \frac{\partial f_n^l \left( \boldsymbol{\Phi}^j(\mathbf{l}) \right)}{\partial \mathbf{l}} \right|_{\mathbf{l}_i} \tag{4}$$

The $\boldsymbol{\omega}$ values now represent the *neurons* which have to be changed the least to affect the upper layer's neuron activation the most, and as such they can be treated as the relevance scores for $f_n^l$ of all lower layer neurons $f^{l-1}$. A large positive $\boldsymbol{\omega}_{n,i,k}^l$ value means the neuron $f_k^{l-1}$ contributes substantially to the activation of $f_n^l$, while a large negative $\boldsymbol{\omega}_{n,i,k}^l$ value inhibits the activation.

The ability to identify the positively and negatively contributing neurons in layer-by-layer fashion is the first step towards understanding the internal representation structure and identifying partially-distributed representations. This can be accomplished by using Step-wise sensitivity analysis and the resulting $\boldsymbol{\omega}^l$ to guide a hierarchical traversal of the network layers. Next, we formally describe our method.

## 3.2 THE METHOD

The main contribution of our algorithm is the granularity with which it can illuminate the internal workings of a DNN. We generate a directed acyclic graph that spans the entire network. We call it a *dependency graph* because it increases the understanding of exactly how the activation of higher output layers depends on lower input layers. It illustrates how the input is transformed in each of the network's layers and highlights "paths" that, when followed, identify the building blocks of higher level features. For example, our method can take us much closer to the ability to say that the concept of a shark is encoded as combination of a fin, body, and tail feature (see Fig. 1a).

Our step-wise sensitivity analysis method is separated into two parts. The first (wrapper) part traverses the network and maintains the input to iteratively apply the second part of the method. The second (step) part applies step-wise sensitivity analysis to identify relevant neurons, as specified in Algorithm 1.

---

**Algorithm 1** Step-wise sensitivity analysis: Identifying partially-distributed representations

---

**INPUT:** DNN classifier $\Phi$, a layer $f^l \in \mathbb{R}^d$ from $\Phi$, a set of relevant neurons $n \in \mathbb{S}$, and a set of images $\mathbf{I}_i \in \mathbb{I}$.

**STEP 1:** Compute relevance of neurons in layer $f^{l-1}$ for each $n$ and $\mathbf{I}_i$ so that if $f^{l-1}$ is a:

    1. Fully-connected layer: stack results into a relevance tensor $\boldsymbol{\omega}_{n,:,:}^l \in \mathbb{R}^{|\mathbb{S}| \times K}$;

    2. Convolutional layer: spatially average the output volume tensor $\boldsymbol{\omega}_{n,i,\dots}^l$ into a relevance tensor $\boldsymbol{\omega}_{n,i,:}^l \in \mathbb{R}^{|\mathbb{S}| \times |\mathbb{I}| \times K}$;

    3. Pooling-layers: directly compute for $l-2$: $\boldsymbol{\omega}^l = \nabla_{f^{l-2}} f^l|_{\mathbf{I}_i}$

**STEP 2:** Select outliers as relevant neurons using $1.5\times$ Inter Quartile Range

**STEP 3:** Rank relevant neurons based on their relevance frequency across images ($\boldsymbol{\omega}_{n,k}^{l,:}$).

**STEP 4:** Select top $b$ relevant neurons, where $b$ is a branching factor.

**OUTPUT:** $b$ relevant neurons for each distinct $n$ in $\mathbb{S}$.

---

The basic idea of our step-wise sensitivity analysis is illustrated in Fig. 1a. Given a DNN classifier $\Phi$ and a set of relevant neurons $n \in \mathbb{S}$, start from the top layer and follow Algorithm 1 to produce a set of $b$ relevant neurons – $\mathbb{B}^n$, for each distinct $n$ in $\mathbb{S}$. Next, set $\mathbb{S}$ to the union of all relevant neurons $\mathbb{B}^n$ ($\mathbb{S} \leftarrow \bigcup_n \mathbb{B}^n$) for the lower layer, and repeat until the input layer. For computational efficiency, the magnitude of $b$ is a threshold for the cardinality of each $\mathbb{B}^n$, thus discarding a proportion of potential relevant neurons. We believe that $b$ is an important hyper-parameter since it limits the size of potential PDRs, which recent studies indicate to be typically between 8 and 50 neurons (Fong & Vedaldi, 2018). We detail each of the steps in Algorithm 1 next.

STEP 1: COMPUTE RELEVANCE TENSOR

**Input:** This step requires a network ($\Phi$), a layer $f^l$, a neuron $n \in f^l$, and an image $\mathbf{I}_i$.

**Output:** Computes the relevance score of neurons in layer $f^l - 1$ with respect to a neuron $n$ in layer $f^l$ as a gradient at $\mathbf{I}_i$ using Equation equation 4. Essentially, this produces the relevance of all neurons in layer $f^l - 1$ to the activation of neuron $n$.

**Method:** The relevance for DCNN is computed differently depending on the type of layer $f^l$.

If $f^l$ is **fully-connected**, the result is a relevance vector $\boldsymbol{\omega}_{n,i,:}^l \in \mathbb{R}^{|f^{l-1}|}$. Repeating this process for all images and neurons in $\mathbb{S}$ yields a relevance tensor $\boldsymbol{\omega}^l$.

If $l$ is a **convolutional** layer, the result of Equation equation 4 is a 5D relevance tensor $\boldsymbol{\omega}_{n,i,\dots}^l \in \mathbb{R}^{H \times W \times K}$, where $H, W, K$ are respectively the height, width, and number of activation maps in $l-1$. Since every activation map $k$ is produced by convolving identical weights onto a lower layer activation map $p$, $k$ represents the existence of an identical feature across $p$. Hence, the vector $\boldsymbol{\omega}_{n,i,h,w,:}^l$ represents the relevance of all lower level activation maps (features) at a location $(h, w)$ to the activation of $n$. Since we are interested in the relative importance of a feature, we perform spatial-averaging over all locations $(h, w)$ to convert $\boldsymbol{\omega}_i^{f_n^l}$ into a relevance vector $\boldsymbol{\omega}_i^{f_n^l} \in \mathbb{R}^K$, where each dimension indicates the relative importance of an activation map across locations. This formulation enables us to repeat the process for all images, neurons and again obtain a 3D relevance tensor $\boldsymbol{\omega}^l$.

The **pooling** layers can be seen as a filter of their predecessors since $\frac{df^l}{d\mathbf{I}_i} = c \times \frac{df^{l-1}}{d\mathbf{I}_i}$, where $c \in \{0, 1\}$. Hence, if $f^{l-1}$ is a pooling layer we compute the relevance tensor directly w.r.t $l-2$: $\boldsymbol{\omega}^l = \nabla_{f^{l-2}} f^l|_{\mathbf{I}_i}$.

STEP 2: OUTLIER DETECTION

**Input:** This steps requires a relevance tensor $\boldsymbol{\omega}^l$.

**Output:** The result identifies neurons relevant for neuron $n$ in layer $l-1$.

**Method:** Our preliminary experiments indicated that each row $\boldsymbol{\omega}_{n,i,:}^l$ follows a normal distribution, and consistently exhibits a small number of outliers across $i$ (see Figure 1b). Therefore, we make two simplifying assumptions. First, we assume these outliers are the only relevant neurons. Second, we choose to focus on only the positive $\boldsymbol{\omega}$ outlier values and leave the analysis of negative outliers

for future work. Finally, we use the Tukey's fences ($1.5\times$ Inter-Quartile Range) outlier detection method (Tukey, 1977) to select relevant neurons from each row $\boldsymbol{\omega}^l_{n,i,:}$. Figure 1b illustrates this step.

STEP 3: RANKING

**Input:** Outliers in $\boldsymbol{\omega}^l_{n,:,:}$.
**Output:** Relevance ranking of each lower-layer neuron.
**Method:** We use the outlier detection procedure to detect relevant neurons in every row. We rank the neurons based on the number of $i$ columns of $\boldsymbol{\omega}^l_{n,:,k}$ in which they appear as relevant, resulting in a relevance ranking for each $k \in f^{l-1}$.

STEP 4: SELECT

**Input:** Relevance ranking for each $k \in f^{l-1}$.
**Output:** $b$ relevant neurons for each distinct $n$ in $\mathbb{S}$.
**Method:** Select top $b$ most frequent neurons $\mathbb{B} \subset \forall_k k \in \boldsymbol{\omega}^l_{n,:,k}$ as relevant, where $b = |\mathbb{B}|$ is a branching factor.

## 4 RESULTS AND DISCUSSION

We now demonstrate how our method can be applied to the ImageNet dataset (Russakovsky et al., 2015) for the 16-layer VGG network (VGG16) (Simonyan & Zisserman, 2014). We use the publicly available pre-trained model implemented in the deep learning framework keras (Chollet et al., 2015) and modify the keras-vis (Kotikalapudi & contributors, 2017) implementation of sensitivity analysis using the Guided-backpropagation algorithm (Springenberg et al., 2014). We perform experiments with 10 classes and select 100 images per class from the training set for which VGG16 the probability mass of the correct class is above 99%. The execution takes approximately 12 hours on NVIDIA Tesla K80 GPU to traverse the entire network for one class with a branching factor of 3. However, most of the time is spent within the convolutional layers, which have a larger number of unique neurons. Since our approach's time complexity can be reduced to worst-case $O(b^d)$, where $b$ is the branching vector and $d$ is the depth ($d = 22$ in the case of VGG16) we constrain $b$ due to computational limitations. The approach is still practical since it is not designed to be executed every time that an explanation is necessary, just as a network is not retrained every time before a prediction. In Section 4.1 we show the particular occurrence of outliers within the $\boldsymbol{\omega}^l$ values to demonstrate how our method chooses relevant neurons.

Furthermore, we provide in Section 4.1 a quantitative justification behind the choice of heuristics to detect relevant outliers, while in Section 4.2 we demonstrate our methodology on two classes, demonstrate that the results generalise across 10 class, and illustrate the importance of statistical interpretability.

The novelty of our approach is that it generates a dependency graph of relevant neurons. In contrast to other approaches, we study the process of feature formation and the interdependence of relevant features by following a path through the graph. This allows us to selectively visualise relevant activation maps through sensitivity analysis. We demonstrate not only an increase in the granularity of the state-of-the-art interpretability capabilities, but also that an analysis on a single image could lead to false assumptions about the operation of the network, as demonstrated in Section 4.2. We use $f^{fc}$ to refer to a fully-connected layer, and $f^{b_i c j}$ to refer to $block_n conv_j$ convolutional layer.

### 4.1 QUANTITATIVE JUSTIFICATION OF OUTLIERS AS RELEVANT NEURONS

According to the $\boldsymbol{\omega}$ values, there is a consistent presence of a small number of relevant outlier neurons (less than 6%) – see Figure 2b. The outlier neurons are not identical for different input stimulus of the same class, as can be seen in Figure 2a. This indicates that our approach is not identical to simply selecting the neurons with highest weights, which would yield constant results across images. On the contrary, the frequency of relevance follows a power-law distribution. This suggests that the most frequently occurring neurons could be the main "drivers" (the most pertinent) for the class activation, while the other relevant neurons pick-up nuances or modulate the main drivers. In other words, the most relevant neurons form a basis, which is transformed by the less

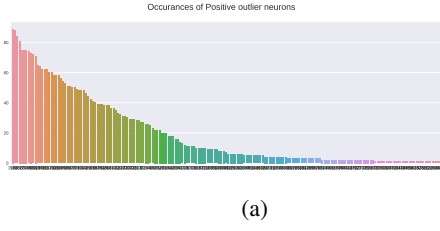
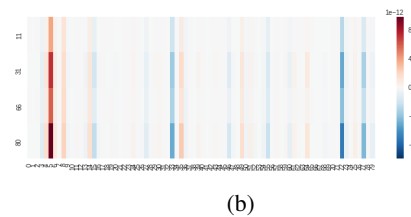

|        (a)        |        (b)        |

Figure 2: (a) Barplot representing the frequency of occurrence for the positive outliers in layer $f^{fc2}$ for the Hammerhead shark class- the y-axis represents the number of images, in which a neuron was a positive outlier. There are 189 unique outliers (4.6% of the total 4096 neurons). Notice that the first 3 outliers occur in almost all images and that the relevance follows a power-law distribution. (b) A heatmap amplified visualisation of outlier neurons, the $\boldsymbol{\omega}^{fc2}_{0:200,i,k}$ values for 4 Hammerhead shark images are cubed. The x-,y-,z-axes represent respectively $k$, $n$, and the number of images where $k$ is relevant for $n$. Observe that the images share exactly the same small number of positive and negative outliers with varying degrees of intensity.

relevant neuron. In some sense, we are finding the inverse of Szegedy et al. (2013)'s adversarial examples – high probability low-dimensional "pockets" in the manifold. What is surprising about Figure 2b is that it is not only consistent with the power law distribution result, but also that different images of the same class share peaks and troughs at exactly the same neurons. We hypothesise that Figure 2b is a visualisation of part of the PDR for a Hammerhead shark in layer $f^{fc2}$. To investigate this hypothesis, we generate dependency graphs of relevant neurons for 10 different classes, which we describe next.

## 4.2 DEPENDENCY GRAPH

Figure 3 illustrates two examples of the output of Step-wise sensitivity analysis with two important observations. First, graphs in Figures 3a & 3b for different classes may share significant similarities. For example, the subgraphs of both Hammerhead shark and Egyptian cat starting from $f^{fc2}_{1820}$ reveal similar most relevant neurons (e.g., $f^{fc2}_{1820}, f^{fc1}_{3116}, f^{fc1}_{2053}$), *identical* subgraph structure (e.g., $f^{fc1}_{3116}$ and its top 3 most relevant neurons) and *similar* subgraph structure (e.g., $f^{fc1}_{2053}$, only one relevant child neuron is shared – $f^{b5c3}_{49}$). Interestingly, the two classes share 6 out of the 8 most relevant activation maps in `block_5_conv3`– 41, 49, 155, 256, 313, 335. This implies that the dependency graphs open the frontier for pattern matching and analysis of network motifs (Milo et al., 2002) across classes. We hypothesise that the emerging network motifs will give a strong indication of the various PDRs positions within a layer and how upper PDRs are leveraging them. In Section 4.2.1 we present further evidence about the existence of network motifs.

Second, Figures 3a & 3b show that both dependency graphs share multiple incoming connections to the very same neuron ($f^{b5c3}_{155}$). It is surprising that this is the first time a neuron is shared within a class. Consequently, the interpretation of the dependency graph enables us to infer an additional relevance metric for a neuron – its inter-connectedness according to the number of incoming edges.

Therefore, step-wise sensitivity analysis allows researchers to focus analysis and interpretation efforts on the most pertinent regions of a DNN. For example, Figures 4b & 4b display targeted visualisation through sensitivity analysis of the especially relevant neuron $f^{b5c3}_{155}$. Had we relied on a single visualisation, we would have erroneously presumed that the neuron perfectly encodes either the idea of a shark or of a cat. However, step-wise sensitivity analysis exposes that the neuron is equally important for both classes, and forms a part of a shared sub-structure. Therefore, it must encode a more abstract concept.

Exploring a neuron or activation map in isolation is simplistic. In reality, the semantics are expressed within the combination of neurons within the PDR. In future work, we will apply activation maximisation Erhan et al. (2009) of an entire PDR to investigate its semantic properties.

### 4.2.1 SHARED DEPENDENCY GRAPHS

In order to investigate the generalisability of the results in Figure 3, we transformed the 10 dependency graphs into bag-of-nodes features representations. Then we performed the ward method (Murtagh & Legendre, 2011) for hierarchical agglomerative clustering with cosine distance similarity to group the dependency graphs.

Interestingly, closer inspection of Figure 4a reveals three clusters of most similar dependency graphs – 1) hammer head and tiger shark; 2) African and Indian elephant; 3) German Sheppard and great white shark. Naturally, the first two clusters consist of the most semantically and visually similar classes – this supports the validity of our approach. Surprisingly, cluster 3) suggests an unnatural similarity between animals. One possible explanation could be that both of these classes share a PDR encoding sharp teeth. Another interesting observation is that the graphs are separated into two general clusters – one consisting of the all the sharks, the German Shepherd, and the Persian cat; and another consisting of the elephants, the Labrador Retriever and the Egyptian cat. One natural semantic separation between the two groups could be the degree of "danger".

Finally, as expected, most of the lower layer activation maps are shared across all classes since they encode very abstract features. At the same time, while there is a large proportion of abstract neurons shared across all dependency graphs, the upper dendrogram in Figure 4a depicts other more specialised neuron clusters, which are idiosyncratic to their semantic groupings.

These three observations support the hypothesis that the dependency graphs reveal semantically meaningful groups of neurons across classes that form PDRs. Identifying and analysing such specific sub-graphs is a non-trivial graph theory problem, which we leave for future work. Once we are able to accurately extract the shared sub-graphs, we will be able to provide hierarchical explanation behind a particular decision. For example, the classification was a shark because the network detected a sea, sharp teeth, a fish tail and a long fin.

## 5 CONCLUSIONS & FUTURE WORK

In this paper we make three contributions to the area of interpreting learned representations. First, we are the first to propose a statistical DNN interpretability method that aggregates results of an instance-specific method to gain model-centric results. Second, we build a dependency graph of the relevant neurons to gain finer-grained understanding of the nature and interactions of a DNN's internal features. Third, we propose three new relevance metrics to identify salient neurons: 1) *the outlier weights* of a linear approximation of the neuron activation; 2) *the ranking score* of a neuron based on its frequency as an outlier across multiple input stimuli; 3) *the interconnectedness* of a neuron within the dependency graph. Our method modifies sensitivity analysis into Step-wise sensitivity analysis that applies the same linear approximation – but on a layer-by-layer basis as opposed to the usual output-input basis. We demonstrate that the results generalise by illuminating shared subgraphs across 10 classes. These subgraphs can be grouped into semantic clusters since they contain the quintessential neurons of a class.

Step-wise sensitivity analysis opens an opportunity for further and more focused explorations of the internal operations of DNNs. Although it can still be applied on a single image to gain instance-specific interpretability, we argue that to gain a statistically viable result it is important to conduct analysis both, across classes and images. In the future, we will apply the approach to facilitate error explanation and decision justification on a much lower level by providing semantic interpretation of the discovered PDRs through visualisation approaches. We will demonstrate the features that make the difference between semantically similar classes and quantify the interpretability of the resulting PDRs using concept segmentation as in (Fong & Vedaldi, 2018; Bau et al., 2017). Further, we will investigate the suitability of our approach to defend against adversarial attacks. Finally, we will explore the possibility to use the dependency graphs to prune the network in order to perform network compression, or the extract binary classifiers for particular classes in the form of dependency graphs to distil the network.

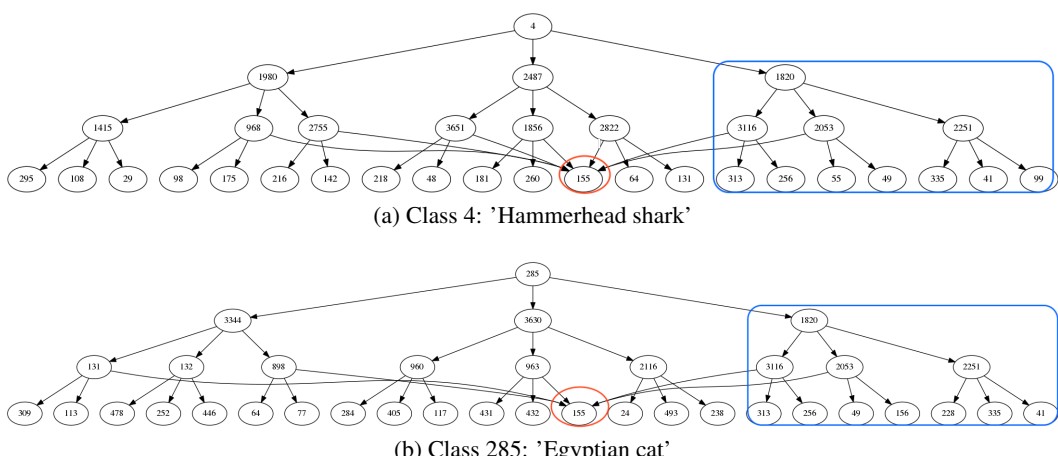

(a) Class 4: 'Hammerhead shark'

(b) Class 285: 'Egyptian cat'

Figure 3: Dependency graphs for hammerhead shark and Egyptian cat classes of the relevant neurons for the penultimate 4 layers, excluding the pooling layer. The graphs expose the links only between the relevant neurons with a branching factor of 3. Notice the multiple connections to $f_{155}^{b5c3}$ (red circle). Notice also the similarities in the subgraphs of $f_{1820}^{fc2}$ (blue rectangle) for both, shark and cat classes.

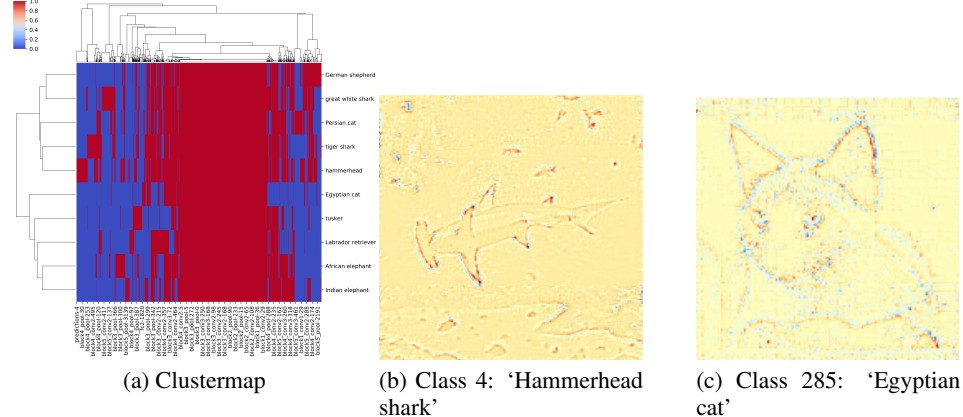

(a) Clustermap

(b) Class 4: 'Hammerhead shark'

(c) Class 285: 'Egyptian cat'

Figure 4: a) A clustered heatmap (clustermap) of each of the 10 dependency graphs (spanning the entire network) into a bag-of-nodes features representation. The x,y,z-axis respectively represent neuron, class, and presence of the neuron in the dependency graph – red present, blue abscent. The dendrograms on the side indicate the relative distance between points and clusters. Notice the three small clusters of semantically similar classes on the side. b) & c) Heatmaps representing standard sensitivity analysis of activation map $f_{155}^{b5c3}$ indicating the regions of the image from the corresponding class. Red and blue respectively correspond to positive or negative contribution to the activation.

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

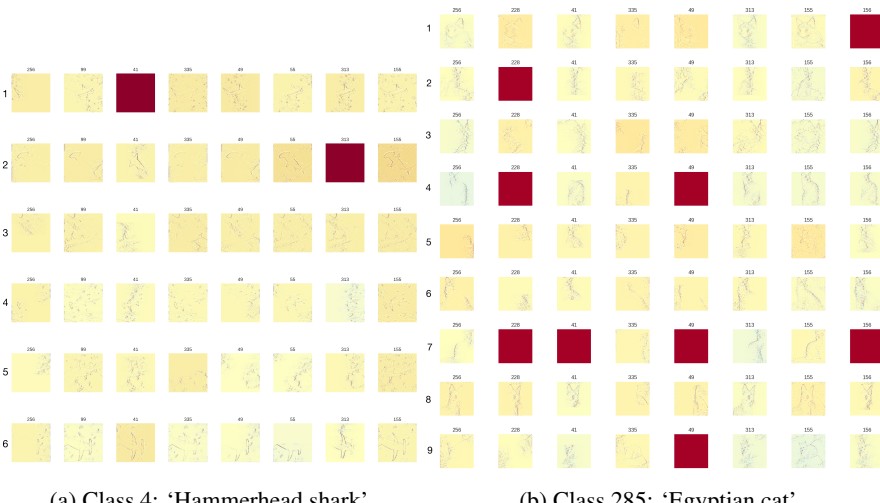

(a) Class 4: 'Hammerhead shark'    (b) Class 285: 'Egyptian cat'

Figure 5: A heatmap of all activation maps at layer $f^{b5c3}$, relevant to neuron $f^{fc2}_{1820}$ for the respective classes. The red heatmaps indicate absence of relevant pixels to a particular activation map (best viewed in digital).

# 6 APPENDIX

## 6.1 INVESTIGATION OF THE SHARED FEATURES

An important claim of our paper is that a comparison of activation maps across images can lead to a better understanding of the effect of a feature. For instance, rows 1 & 5, column 256 in Figure 5a could lead to the erroneous conclusion that $f^{b5c3}_{256}$ detects shark tails, while it also activated for the shark head and also for front and rear parts of cats.

Figure 5b suggests that neurons $f^{b5c3}_{49}$ and $f^{b5c3}_{335}$ are complementary. They activate for similar regions; however, they each capture different parts (e.g., a cat's ear – row 1; the region of $f^{b5c3}_{335}$ is contained within that of $f^{b5c3}_{49}$, but has a much sharper boundary). In future work we will explore the exact relationship between such neurons.

