# OpenReview forum: "Step-wise Sensitivity Analysis: Identifying Partially Distributed Representations for Interpretable Deep Learning"
_ICLR.cc/2019/Conference_

### Official Review · AnonReviewer1 · 2018-11-02
**Interesting idea, but serious concerns about whether it produces meaningful results**

**Rating:** 3
**Confidence:** 4

**Review:**

Summary:
The paper introduces a new approach for interpreting deep neural networks called step-wise sensitivity analysis. The approach is conceptually quite simple and involves some interesting ideas, but I have some serious concerns whether the output produced by this method carries any meaning at all. If the authors were able to refute my concerns detailed below, I would raise my score substantially.


Strengths:
+ Potentially interesting heuristic to identify groups of feature channels in DNNs that encode image features in a distributed way


Weaknesses:
- Using the magnitude of the gradient in intermediate layers of ReLU networks is not indicative of importance
- No verification of the method on a simple toy example


Details:


Main issue: Magnitude of the gradient as a measure of importance.

I have trouble with the use of the gradient to identify "outliers," which are deemed important. Comparing the magnitude of activations across features does not make sense in a convnet with ReLUs, because the scale of activations in each feature map is arbitrary and meaningless. Consider a feature map h^l[i,x,y,f] (l=layer, i=images, x/y=pixels, f=feature channels), convolution kernels w^l[x,y,k,f] (k=input channels, f=output channels) and biases b^l[f]:

h^l[i,:,:,f] = ReLU(b^l[f] + \sum_k h^(l-1)[i,:,:,k] * w^l[:,:,k,f])

Assume, without loss of generality, the feature map h^l[:,:,:,f] has mean zero and unit variance, computed over all images (i) in the training set and all pixels (x,y). Let's multiply all "incoming" convolution kernels w^l[:,:,:,f] and biases b^l[f] by 10. As a result, this feature map will now have a variance of 100 (over images and pixels). Additionally, let's divide all "outgoing" kernels w^(l+1)[:,:,f,:] by 10.

Simple linear algebra suffices to verify that the next layer's features h^(l+1) -- and therefore the entire network output -- are unaffected by this manipulation. However, the gradient of all units in this feature map is 10x as high as that of the original network. Of course the gradient in layer l-1 will be unaltered once we backpropagate through w^l, but because of the authors' selection of "outlier" units, their graph will look vastly different.

In other words, it is unclear to me how any method based on gradients should be able to meaningfully assign "importance" to entire feature maps. One could potentially start with the assumption of equal importance when averaged over all images in the dataset and normalize the activations. For instance, ReLU networks with batch norm and without post-normalization scaling would satisfy this assumption. However, for VGG-16 studied here, this is not the case.

On a related note, the authors' observation in Fig. 4b that the same features are both strongly positive and strongly negative outliers for the same class suggests that this feature simply has a higher variance than the others in the same layer and is therefore picked most of the time. Similarly, the fact that vastly different classes such as shark and German Sheppard share the same subgraphs speaks to the same potential issue.


Secondary issue: No verification of the method on simple, understandable toy example.

As shown by Kindermans et al. [1], gradient-based attribution methods fail to produce the correct result even for the simplest possible linear examples. The authors do not seem to be aware of this work (at least it's not cited), so I suggest they have a look and discuss the implications w.r.t. their own work. In addition, I think the authors should demonstrate on a simple, controlled (e.g. linear) toy example that their method works as expected before jumping to a deep neural network. I suppose the issue discussed above will also surface in purely linear multi-layer networks, where the intermediate layers (and their gradients) can be rescaled arbitrarily without changing the network's function.


References:
[1] Kindermans P-J, Schütt KT, Alber M, Müller K-R, Erhan D, Kim B, Dähne S (2017) Learning how to explain neural networks: PatternNet and PatternAttribution. arXiv:170505598. Available at: http://arxiv.org/abs/1705.05598

---

### Official Review · AnonReviewer3 · 2018-11-04
**Ok but not good enough**

**Rating:** 4
**Confidence:** 4

**Review:**

Summary:
This paper introduces step-wise sensitivity analysis (SSA), which is a modification of saliency maps (Baehrens et al. 2010, Simonyan et al. 2013) to a per-layer implementation. Instead of only measuring the importance of input nodes (e.g. pixels) to the classification, SSA measures the importance of all nodes at each layer. This allows for a way to find the important sub-nodes for each node in the tree given a particular sample. It is then straightforward to aggregate results across different input samples and output a dependency graph for nodes.

Novelty:
The technical contribution is a very simple extension of Simonyan et al. 2013. The main novelty lies within the created dependency graph from the node importance weights, but the usefulness of such graph is unclear. In addition, the claim that this is the first method that aggregates results of an instance-specific method to gain model-centric results is a stretch considering other works have found important nodes or filters for a specific class by aggregating across instance-specific samples (Yosinski et al. 2015).

Evaluation:
The idea of producing an interpretable dependency graph for nodes is interesting, and the possible conclusions from such graphs seem promising. However, most of the interesting possible conclusions seem to be put off for future work. I don’t believe the experiments are sufficient to show the significance of SSA. The main hypothesis is that dependency graphs allow for a way to interpret the model across samples, but it doesn’t show any conclusive results about the data or models that wasn’t previously known. The results are mostly speculative, such as the fact that German shepherd and great white shark nodes are clustered together, possibly due to the fact that both of these classes share a PDR encoding sharp teeth, but that is never actually demonstrated.

---

### Official Review · AnonReviewer2 · 2018-11-08
**Replacement for an original reviewer**

**Rating:** 3
**Confidence:** 5

**Review:**

The paper proposes a modification of the saliency map/gradient approach to explain neural networks.

# Method summary

The approach is as follows:
For each layer, the gradient w.r.t. it's input layer is computed for multiple images concurrently.
Then for conv layers, the activations are averaged per feature map (over space).
As a result, for both fully connected and convolutional layers there is a 3D feature map.
From these at most b positive outliers are selected to be propagated further.
What is a bit strange is that in the results section, guided backpropagation is mentioned and clearly used in the visualizations but not mentioned in the technical description.

# Recommendation

The current evaluation is definitely not sufficient for acceptance.
The evaluation is done in a purely qualitative matter (even in section 4.1 Quantitive justification of outliers as relevant neurons). The results appear to be interesting but there is no effort done to confirm that the neurons considered to be relevant are truly relevant. On top of that, it is also evaluated only on a single network and no theoretical justification is provided.

# Discussion w.r.t. the evaluation

To improve section 4.1,  the authors could for example drop out the most important neurons and re-evaluate the model to see whether the selected neurons have a larger impact than randomly selected neurons. Since the network is trained with dropout, it should be somewhat robust to this. This would not be a definitive test, but it would be more convincing than the current evaluation. Furthermore high values do not imply importance.

It might be possible that I misunderstood the experiment in Figure 2. So please correct me if this is the case in the reasoning below.
In figure 2, FC2 is analyzed. This is the second to last layer. So I assume that only the back-propagation from logits (I make this assumption since this is what is done commonly and it is not specified in the paper) to FC2 was used. Since we start at the same output neuron for a single class, all visualisations will use the same weight vector that is propagated back. The only difference between images comes from which Relu's were active but the amount if variability is probably small since the images were selected to be classified with high confidence. Hence, the outliers originate from a large weight to a specific neuron.

The interpretation in the second paragraph of section 4.2.1 is not scientific at all. I looked at the German Shepherd images and there are no teeth visible. But again, this is a claim that can be falsified easily. Compare the results when german Shepherds with teeth visible are used and when they are not. The same holds for the hypothesis of the degree of danger w.r.t. the separation.

Finally, there is no proof that the approach works better than using the magnitude of neuron activations themselves, which would be an interesting baseline.

Additional remarks
---------------------------

The following is an odd formulation since it takes a 3D tensor out of a 5D one and mixes these in the explanation:
"... the result of equation for is a 5D relevance tensor $\omega^l_{n,i,..} \in R^{H\times W\times K} ....."

The quality of the figures is particularly poor.
- Figure 1 b did not help me to understand the concept.
- Figure 2 The text on the figure is unreadable.
- Figure 4a is not readable when printed.

---

### Author Response · Authors · 2018-11-21
**Thank you for the constructive suggestion. A revision of the paper will follow shortly.**

Dear Reviewers,

Our sincerest gratitude for your fruitful comments and thoughtful suggestions!
You have considerably helped us to improve our work.

We have been conducting experiments to address the two main concerns, namely 1) the mathematical grounding of SSA and 2) a stronger evaluation section that contains a simple toy example and demonstrates the utility of the dependency graphs.

A revision of the paper will follow shortly, where we will also include discussions w.r.t  Kindermans et al. (2017) & (Yosinski et al. 2015).

---

### Author Response · Authors · 2018-11-28
**General Rebuttal**

We thank the reviewers for their thoughtful comments.

The main concern of the reviewers is whether the magnitude of gradients can be used to determine neuron relevance and they suggested to illustrate the validity of this approach through a toy example.

First, the novelty of our work is the new approach of generating dependency graphs to explaining neural networks, and we present our first algorithms for computing such graphs. There could be others, which we will explore in future work. We thank Reviewer 1 for bring up PatternNet(Kindermans et al. 2017), which addresses the limitation of gradient-based approaches that in the context of heatmap visualisation the gradients answer the question which pixels makes this classification more or less like bird for example, not which pixels make this classification a bird.  Gradients indicate which neurons should change the least, so that there would be the greatest change in the output (Samek 2017). Since we are interested in the global understanding of the network and not in why in particular this image was classified as a bird, we direct our attention to the neurons that mostly influence the activation of an adjacent neuron.

In the future it would be interesting to compare the quality of dependency graphs produced through sensitivity analysis and PatternNet. However, we believe that the magnitude of the gradients approach is suitable for the current iteration of the paper for two reasons.
First, it is important to clarify that we are talking about the relative magnitudes of the gradients at every neuron, not the absolute. In that case regardless of how much we scale the weights, for every neuron we will have a relative ranking between neurons indicating the sensitivity of the upper layer neuron (target neuron) to its lower layer connections. The ranking function can be improved. The Taylor expansion gives a linear approximation of the target neuron’s activation, the weights of which we use as a proxy for the relevance. We utilise the relative values of these weights separately for every neuron. Hence, a strongly connected upper-layer neuron cannot dominate the relevance ranking of other upper layer neurons.

Second, in a toy example that will appear in the revised version, we compare the averaged (across target neurons) relevance ranking that is computed through activations, gradients and weights. Our experiments demonstrate that on a global level removing the neurons with the strongest absolute weights results in the greatest performance degradation. However, such relevance ranking corresponds solely to the overall performance of the network in contrast to a local performance for a specific class. Due to the fact that the aim of this paper is to extract dependency graphs local to different classes we utilise the gradients strategy, for which on the global level our experiment confirms the supposition that it outperforms the activations strategy.

Our dependency graph evaluation on the large scale VGG16, which computes the performance difference between a forward pass through the proposed dependency graphs and a random dependency graph, would take us longer to complete due to lack of computational resources. Do the reviewers believe it is an integral part of the evaluation section?

Additional minor corrections:

We thank Reviewer 3 for bringing up Yosinski et al. (2015). Although there have been many papers [e.g. Zintgraf (2017); Zhou (2015)], which explore multiple instance-specific results to gain a model-centric understanding, they undertake a manual investigation of the instance-specific results. In contrast, we leverage the output of instance-specific results within our algorithm to produce a model-centric explanation.

We thank Reviewer 2 for pointing out the visualisation approach. We will make it explicit that we leverage guided back-propagation for the visualisations, and we are planing to compare the effect of using the guided-backpropagation definition to compute the relative relevance ranking; however, we have not mentioned guided-backpropagation within the method description because our method is inherently different —  in contrast to guided-backpropagation, we do not necessarily exclude units with negative forward activations.

We will remove the speculative claim about the German Shepard from the evaluation.

It will be made explicit that logits are used at the softmax layer.

References:
Samek, Wojciech, et al. "Evaluating the visualization of what a deep neural network has learned." IEEE transactions on neural networks and learning systems 28.11 (2017): 2660-2673.
Kindermans, Pieter-Jan, et al. "PatternNet and PatternLRP–improving the interpretability of neural networks." stat 1050 (2017): 16.
Zhou, Bolei, et al. "Object detectors emerge in deep scene cnns." (2015).
Zintgraf, Luisa M., et al. "Visualizing deep neural network decisions: Prediction difference analysis." arXiv preprint arXiv:1702.04595 (2017).

---

### Meta-Review · Area_Chair1 · 2018-12-15

**Confidence:** 5
**Recommendation:** Reject

**Metareview:**

This work proposes a modification of gradient based saliency map methods that measure the importance of all nodes at each layer. The reviewers found the novelty is rather marginal and that the evaluation is not up to par (since it's mostly qualitative). The reviewers are in strong agreement that this work does not pass the bar for acceptance.